# m6A Methylation Analysis Reveals Networks and Key Genes Underlying the Coarse and Fine Wool Traits in a Full-sib Merino Family

**DOI:** 10.3390/biology11111637

**Published:** 2022-11-09

**Authors:** Guoying Hua, Xue Yang, Yuhao Ma, Tun Li, Jiankui Wang, Xuemei Deng

**Affiliations:** Key Laboratory of Animal Genetics, Breeding, and Reproduction of the Ministry of Agriculture, China & Beijing Key Laboratory of Animal Genetic Improvement, China Agricultural University, Beijing 100193, China

**Keywords:** MeRIP-seq, wool follicle, m6A modification, PI3K/AKT pathway

## Abstract

**Simple Summary:**

Artificial breeding makes traits move forward in one direction and reach the extreme, such as ultra-fine wool covering the whole body of fine wool sheep. Nevertheless, many other domestic sheep remain the coarse wool type, and some mendelian genome loci have been identified as having major genes for these traits; however, the epigenetic regulation is still unclear.

**Abstract:**

In our study, a set of lambs with coarse wool type all over their bodies were discovered within a full-sib family during an embryo transfer experiment of merino fine wool sheep. The difference between coarse and fine wool traits were studied from the perspective of RNA modification-N6-methyladenosine. A total of 31,153 peaks were collected, including 15,968 peaks in coarse skin samples and 15,185 peaks in fine skin samples. In addition, 7208 genes were differentially m6A methylated, including 4167 upregulated and 3041 downregulated in coarse skin samples. Four key genes (*EDAR*, *FGF5*, *TCHH*, *KRT2*) were obtained by comprehensive analysis of the MeRIP-seq and RNA sequence, which are closely related to primary wool follicle morphogenesis and development. The PI3K/AKT pathway was enriched through different m6A-related genes. These results provided new insights to understand the role of epigenetics in wool sheep domestication and breeding.

## 1. Introduction

N6-methyladenosine (m6A) is a type of RNA modification which affects RNA stability, nuclear transport, splicing, and translation [1]. m6A was first found in poly(A) of RNA fractions in 1974 [2,3]; however, until the 1970s, m6A was considered a mode of RNA regulation because of a lack of study methods for detecting m6A sites. m6A became famous in 2012 when MeRIP-seq was introduced, which was a next-generation sequencing method to detect m6A site by transcriptome [4]. Research found that m6A is usually located near to stop codons of 3′ UTR, and the level of m6A is always changing in response to individual development and environment stress [5]. Thus, m6A plays an important role in RNA function and fate.

Sheep can produce wool fibers for clothing and textiles. Different sheep breeds have different wool characteristics, and there are great differences in fiber diameter, crimp, and elasticity [6]. Chinese coarse wool sheep breeds, such as Mongolian sheep [7], Tibetan sheep [8], and Kazak sheep [9] produce thick medullated wool developed from primary wool follicles. The fine wool sheep breeds in China were originally produced by the crossbreeding of local coarse wool sheep with imported fine wool Merino sheep, such as the former Soviet Merino sheep [10], Australian Merino sheep [11], and Rambouillet sheep [12].

In the merino, therefore, as a result of long-time selective breeding, the woolly coat has been almost exclusively developed, and today it forms homogeneous fleeces covering the whole body which are produced by primary and secondary wool follicles [13]. However, various amounts of kemp fiber mixed with wool (heterogeneous fleece) can be observed occasionally in some newborn merinos, such a fleece being produced by primary wool follicle with the characteristics: coarse, white, opaque, and dull. Shortly after birth, the kemp fibers gradually fall off, and the covering is then composed of woolly fleece only [14]. In this study, an extreme case was observed—four lambs with heterogeneous fleece all over their bodies were discovered within a full-sib family during the embryo transfer experiment of merino fine wool sheep. Meanwhile, their three siblings have the same typical homogeneous fleece as modern merino. In this study, we used these two groups of sheep to analyze the effect of m6A-methylation on early wool follicle development in sheep.

## 2. Materials and Methods

### 2.1. Sample Collection

The one-month-old merino lambs with coarse wool were found in a merino fine wool lamb population during an embryo transfer experiment of Chinese Merino in Chifeng city, Inner Mongolia, China. Four coarse wool and three fine wool full-sib sheep with same gender (female) (Figure 1) were sampled. They were subjected to the same growth environment and feed conditions. At one month of age, about 1 cm^2^ of middle-side dorsal skin of sheep was quickly collected, and immediately frozen in liquid nitrogen for total RNA extraction. After RNA extraction, the total RNA of each sample was divided into two pools, including coarse and fine for m6A immunoprecipitation (IP). Each pool contained three mixed total RNA. Six original extractions from each pool were collected for quantitative Real-Time PCR (qRT-PCR).

### 2.2. Experimental Procedure

Total RNA was extracted from skin tissue (samples same as transcriptome sequencing) using TRIzol Reagent (Invitrogen). After RNA extraction, DNaseI was used to remove residual gDNA. RNA quality was examined by Nanodrop onec spectrophotometer (Thermo Fisher Scientific, Waltham, MA, USA), and RNA integrity was visualized by 1.0% agarose gel electrophoresis. Qualified RNA was finally quantified by Qubit3.0 with Qubit RNA Broad Range Assay kit (Life Technologies, Carlsbad, CA, USA). Subsequently, 50 μg total RNA of each sample was calculated for polyadenylated RNA enrichment by using VAHTS mRNA Capture Beads (VAHTS). This was followed by 20 mM ZnCl_2_ added to the RNA tube, mixed well, and incubated at 95 °C for 5–10 min until the RNA fragments were mainly distributed in 100~200 nt. Then, 10% RNA fragments were saved as “Input” and the rest was used for m6A immunoprecipitation (IP). The specific anti-m6A antibody (Synaptic Systems) was applied for m6A IP. After that, the enriched RNA was extracted by using TRIzol reagent (Invitrogen). The stranded RNA sequencing library was constructed by KC-Digital Stranded mRNA Library Prep Kit for Illumina (Seqhealth), as per the manufacturer’s instruction. The library products of 200–500 nt were enriched and quantified, and finally sequenced on Novaseq 6000 sequencer (Illumina) with PE150 model.

### 2.3. Bioinformatic Analysis Process

Clean reads were harvested by filtering low-quality reads and removing adaptor sequence. Clean reads were further treated with in-house scripts to eliminate duplication bias introduced in library preparation and sequencing. In brief, clean reads were first clustered according to the UMI sequences, in which reads with the same UMI sequence were grouped into the same cluster. Reads in the same cluster were compared to each other by pairwise alignment, and then reads with sequence identity over 95% were extracted to a new sub-cluster. After all sub-clusters were generated, a multiple sequence alignment was performed to get one consensus sequence for each sub-cluster. After these steps, any errors and biases introduced by PCR amplification or sequencing were eliminated. The deduplicated consensus sequences were used for m6A site analysis. They were mapped to the reference genome of Oar_rambouillet_v1.0 from NCBI database using STAR software (version 2.5.3a) (Cold Spring Harbor Laboratory, Cold Spring Harbor, NY, USA) with default parameters [15]. The exomePeak(Version 3.8) software (Xi’an Jiaotong-Liverpool University, Suzhou, China; University of Texas at San Antonio, San Antonio, TX, USA) was used for peak calling [16]. The m6A peaks were annotated using bedtools(Version 2.25.0) (University of Virginia, Charlottesville, VA, USA) [17]. The deepTools(version 2.4.1) (Max Planck Institute of Immunobiology and Epigenetics, Freiburg, Germany) was used for peak distribution analysis [18]. The differential m6A peaks were identified by a python script, using fisher test. Sequence motifs enriched in m6A peak regions were identified using Homer (version 4.10) (University of California, San Diego, CA, USA) [19].

### 2.4. Quantitative Real-Time PCR Validation

A total of four differently m6A methylated genes (EDAR, KRT2, FGF5 and TCHH) were selected for qRT-PCR, which was performed with SYBR Premix ExTaq (Takara, Dalian, China) on ABI 7500 Real-Time PCR System (Applied Biosystems, Foster City, CA, USA). The cDNA synthesis of coding genes was performed using Quantscript RT Kit Quant cDNA (Tiangen Biotech, Beijing, China) with approximately 500 ng of total RNA as the template; SYBR Premix ExTaq (Tiangen, Beijing, China) was used for real-time PCR, which was available for an ABI7500 Real-Time PCR System (Applied Biosystems). The Tm values of each reaction were list in Appendix A. The β-actin gene was used as a reference control. Each plate was repeated three times in independent runs for all references. Gene expression was evaluated by the 2^−∆∆Ct^ method [20].

## 3. Results

### 3.1. DATA Quality Control

In order to remove adaptor and low-quality data, raw data of IP and input were trimmed by perl and cutadapt. Then, the clean data were available. A total of 65,836,752 and 65,814,440 raw data were obtained from coarse and fine skin samples, respectively, including 60,538,940 and 61,402,346 clean reads, respectively. The effective clean reads accounted for 66.83 and 65.06, respectively. In the input library (RNA-seq library), 59,483,600 and 66,350,948 raw data were obtained from coarse and fine skin samples, respectively, including 52,447,544 and 59,261,076 clean reads, respectively, which accounted for 75.24% and 74.39%, respectively. The relative results are presented in Table 1.

### 3.2. Mapping Reads to the Reference Genome

The clean reads were mapped to sheep genome (Oar_rambouillet_v1.0). The results were as follows: the mapping ratios of m6A-seq library between IP samples of coarse and fine were 96.52% and 96.29%, respectively, while the mapping ratios of RNA-seq library between coarse and fine input samples were 96.03% and 96.04%, respectively. More details are shown in Table 2. The distribution statistics of mapped reads located on the reference genome were used to detect the source localization regions on the sequencing genome, which include CDS, intergenic, and intron region. In general, the percentage of CDS region reads should be the highest. In this study, the percentage of mapped reads in exons between coarse and fine samples (IP and Input) was much higher than the percentage in intergenic and intron regions, which is shown in Figure 2.

### 3.3. m6A Peak Calling and Differential Methylation Analysis

In order to find the m6A modification sites, we divided the whole genome into 20 bp segments, then counted the reads of IP and Input samples, which are within these ranges [16]. The fisher test was performed through total reads. If IP group was significantly higher than Input group (*p* < 0.05), it meant that the region was bound by protein. Then, the adjacent region was merged together as the final protein binding sites. That would be the final peaks. A total of 15,968 peaks were obtained in coarse skin samples, while 15,185 peaks were obtained in fine skin samples (Appendix A). We combined IP and Input samples to indicate the enrichment of reads near to the transcriptome initiation site of the genes. The peak distribution of the peak genes around the gene functional region are shown in Figure 3A. The ChIPseeker was used to analyze the difference peaks genes [21]. P-value of less than 0.05 was used as a threshold of differential peak genes. A total of 7208 differential peaks genes were detected, of which 4167 m6A peaks related genes were upregulated and 3041 peaks genes were downregulated in coarse group (Appendix A). Especially, we found 1711 specific m6A in coarse wool group and 1337 specific m6A in fine wool group (Appendix A). The transcripts were divided into four regions, namely 5′UTR (5′ untranslated region), 3′UTR, CDS (Coding sequence), and NP_exon (Non-protein_exon) (Figure 3B). The differential m6A peaks were mainly enriched in the 3′UTRs, which are shown in Figure 3B.

### 3.4. Motif Analysis

Motif plays an important role for the binding between RNA and protein [22]. Different motifs have different characteristics, and the binding ability and types of proteins are also different [23]. Therefore, we used Homer to extract the peaks of the range for scanning the common motif among peaks and draw motif map, which includes the width, p-value, ratio of targets, and background of each motif (Figure 4). The reported RNA motif structures are characterized by RRACH (R = A or G, H = A, C or U) and the differential m6A peaks in this study were also GGACU motif [24], which is the top rank motif and also the typical m6A motif.

### 3.5. Association Analysis between m6A Modification and Genes Expression

RPKM (reads per kilobase of exon model per million mapped reads) was used to calculate the gene expression level in coarse and fine wool skin samples [25]. A total of 21,954 genes were identified, 4856 genes were considered as the differentially expressed genes (Foldchange > 2 or <0.5). Among them, 2220 were upregulated and 2636 were downregulated in coarse skin sample. The differently expressed genes are listed in Appendix A. Gene expression density map and box plot are shown in Figure 5A,B. On the other hand, in the MeRIP-seq results, the peak abundance changes were used to identify different methylated genes. In this study, we compared the transcriptional level with the methylation level by conducting a correlation analysis of these two omics. The results showed that a total of 300 genes overlapped between upexpressed genes and upmethylated genes in coarse skin sample, while 195 genes overlapped between upexpressed genes and specific methylated genes in coarse skin sample (Figure 5C). Moreover, the above two overlapped genes also share 193 genes in common (Figure 5C). Interestingly, the EDAR (Ectodysplasin A Receptor) gene was included, which was an important gene in controlling primary hair follicle development [26]. In addition, the results also show a total of 237 overlapped genes between downexpressed genes and upmethylated genes in coarse skin sample, while also showing 113 overlapped genes between downexpressed genes and specific methylated genes in coarse skin sample (Figure 5D). Also, the above two overlapped genes shared 103 genes in common (Figure 5D). FGF5 (Fibroblast Growth Factor 5) and TCHH (Trichohyalin) were included, which may affect hair curl in most/all world populations, while other genes such as EDAR and WNT10A (Wnt Family Member 10A) only affect specific populations [27].

### 3.6. GO Analysis and KEGG Pathway Analysis of Differentially Methylated Genes

To analyze the biological function of m6A modified genes, GO (Gene Ontology, http://www.geneontology.org, accessed on 5 January 2021) and KEGG (http://www.genome.jp/kegg/, accessed on 5 January 2021) databases were used to explore the related pathways of m6A modified genes [28]. The top 17 in biological process, top 8 in cellular component, and top 7 in molecular function are shown in Figure 6A. KEGG pathway analysis showed that up-m6A modified genes and specific-m6A modified genes were significantly associated with PI3K-AKT pathway (Figure 6B,C).

### 3.7. Validation of Candidate Genes by qPCR

In this study, we identified some important candidate genes that were related to coarse wool development, such as EDAR, FGF5, and TCHH. Q-PCR was used to verify gene expression differences between coarse and fine wool sheep skin samples. The result of Q-PCR analysis for four differentially expressed genes were consistent with those of RNA-seq, according to which EDAR and KRT2 were upexpressed in coarse skin samples, while FGF5 and TCHH were downexpressed in coarse skin samples (Figure 7).

## 4. Discussion

In the phenotypic analysis, we found that, although the coarse and fine wool lambs are siblings from the same family, they had different wool characteristics, one group presenting heterogeneous wool characteristics and the other presenting homogenous wool characteristics. The difference between the above two groups is the presence or absence of medullated wool developed from primary wool follicles [29]. This indicates that the primary wool follicle is the main basis for wool type for producing coarse wool phenotype. We used these two groups of sheep to analyze the effect of m6A-56 methylation on early wool follicle development in sheep. Therefore, the research objective of this study was to explore the molecular mechanism of primary wool follicle development.

FGF5 gene is a well-known gene that affects hair length, and is closely related to hair follicle development and hair shaft length in mammals, such as humans [30], cats [31], and sheep [32]. Our model showed that the m6A level of FGF5 gene in the coarse wool group was significantly higher than that in the fine wool group, while its expression level was significantly lower than that in the fine wool group, which may be a direct reason for the longer wool of coarse wool sheep compared to that of fine wool sheep. Besides, the coarse wool of sheep was characterized by an outer coat with coarse, white, straight appearance and an under coat with crimped, finer, and closely arranged hair on the body surface [33]. However, fine wool sheep only have curly wool in the inner layer. Therefore, the other phenotype difference between the two kinds of sheep lies in the waviness of wool. Researchers have found that TCHH gene significantly affect the hair waviness [34]. Our detection showed that TCHH gene was more highly expressed in the fine wool group, which may be a molecular marker of the curly trait in the fine wool group. In addition, EDAR regulates the development of primary hair follicles, and EDAR is the major gene associated with the difference of curly trait between Asians and Europeans [35]. Our results showed that TCHH gene was more highly expressed in the fine wool group, which may be a molecular reason of the curly trait in the fine wool group. We found that PI3K/AKT signaling pathway (Figure 8) was significantly enriched in m6A-related genes. Recent studies have found that PI3K/AKT signaling pathway is essential for hair follicle development [36], and whether PI3K/AKT signaling pathway could regulate the occurrence and development of primary hair follicles remains to be further studied.

## 5. Conclusions

Overall, in this study, fine and coarse wool sheep were used to study the molecular mechanism of wool follicle development. From the perspective of mRNA modification, EDAR, TCHH, and FGF5 genes, as well as PI3K/AKT signaling pathway were firstly found to be related to the generation of coarse hair phenotype. We provide a new perspective for studying the role of epigenetics in the domestication and breeding of fine wool sheep.

## Figures and Tables

**Figure 1 biology-11-01637-f001:**
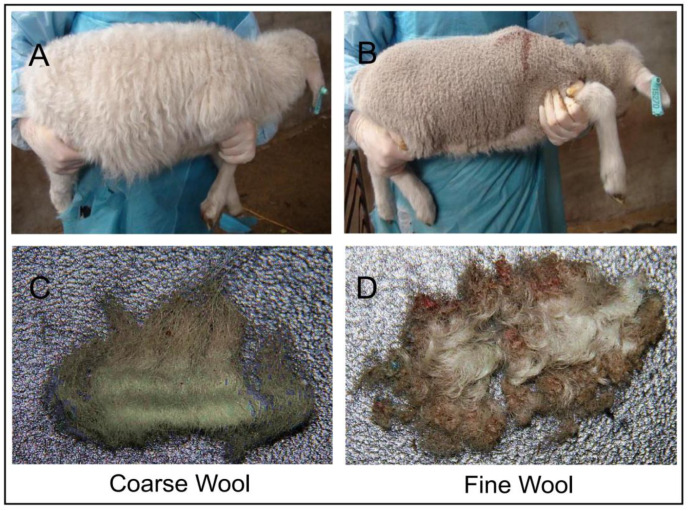
Coarse wool and fine wool merino sheep, showing the characteristic wool traits. (**A**) Appearance of coarse wool lamb. (**B**) Appearance of fine wool lamb. (**C**) Coarse wool fibers from coarse wool lambs. (**D**) Fine wool fibers from fine wool lambs.

**Figure 2 biology-11-01637-f002:**
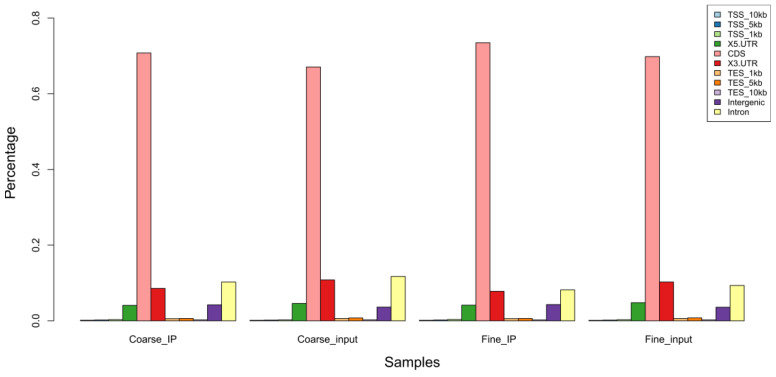
Reads distribution across genomic regions.

**Figure 3 biology-11-01637-f003:**
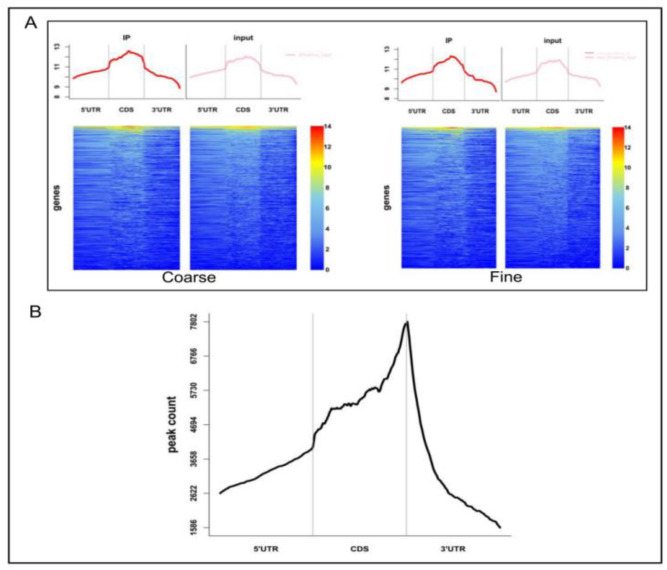
Heatmap of IP and Input reads. (**A**) Heatmap of IP and Input reads at the transcriptome initiation site of the genes. (**B**) Distribution of differential peaks on gene functional regions.

**Figure 4 biology-11-01637-f004:**
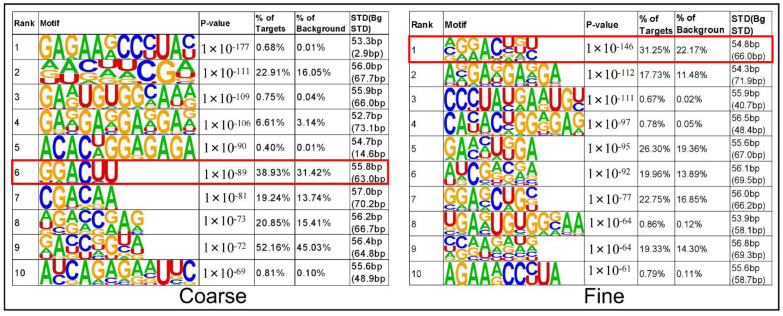
The top 10 motif logos were identified from differential m6A peaks.

**Figure 5 biology-11-01637-f005:**
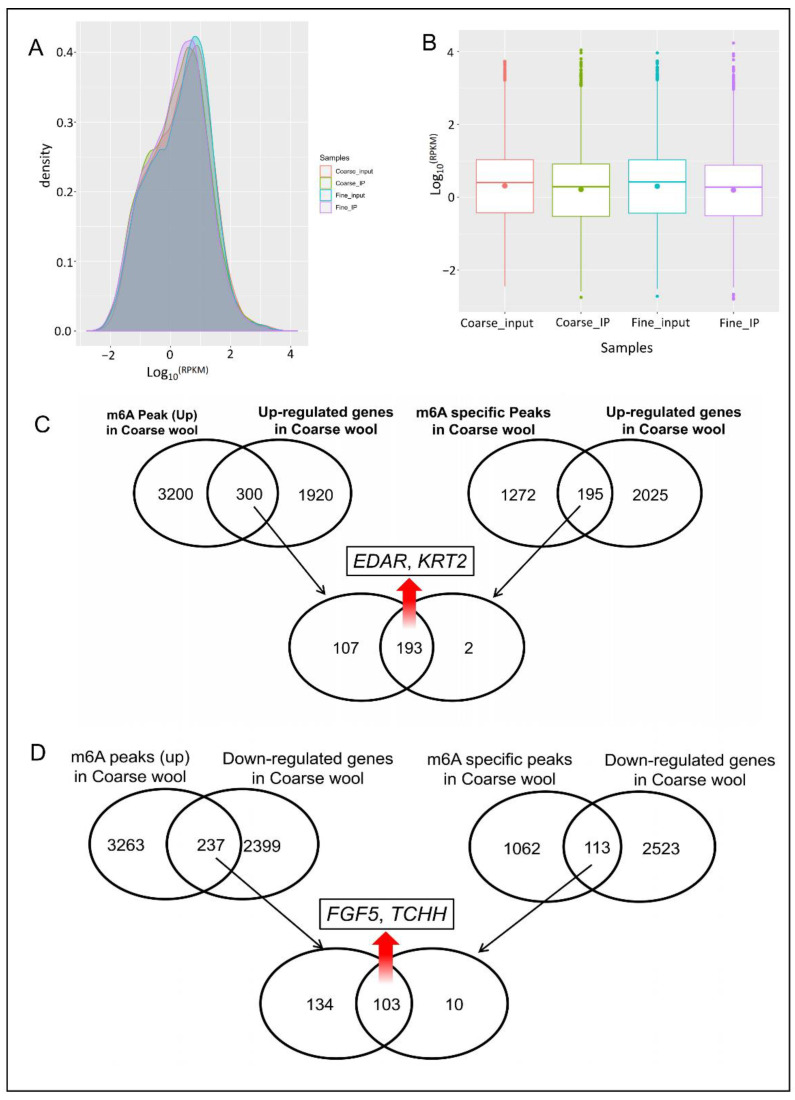
Summary of differentially expressed genes in coarse and fine skin samples. (**A**) Gene expression density map. (**B**) Gene expression box plot. (**C**) Overlapped analysis of up m6A peaks, m6A specific peaks, and upregulated genes in coarse skin samples. (**D**) Overlapped analysis of up m6A peaks, m6A specific peaks, and downregulated genes in coarse skin samples.

**Figure 6 biology-11-01637-f006:**
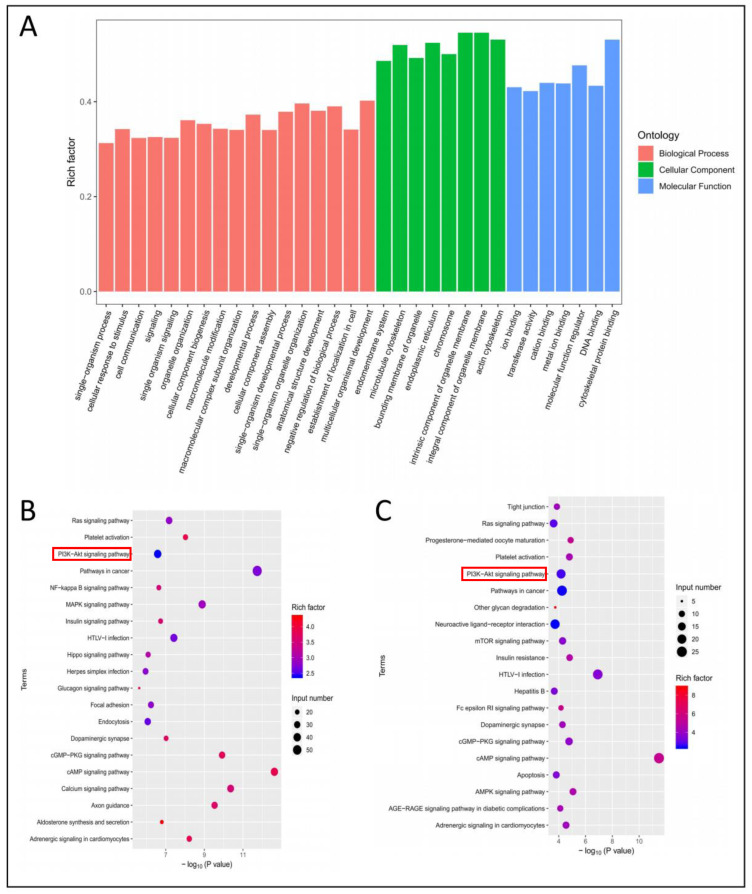
Gene ontology enrichment and KEGG analysis of differential m6A modified genes. (**A**) GO terms of m6A modified genes. (**B**) The top 20 significant pathways of up-m6A modified genes in coarse skin samples. (**C**) The top 20 significant pathways of specific-m6A modified genes in coarse skin samples.

**Figure 7 biology-11-01637-f007:**
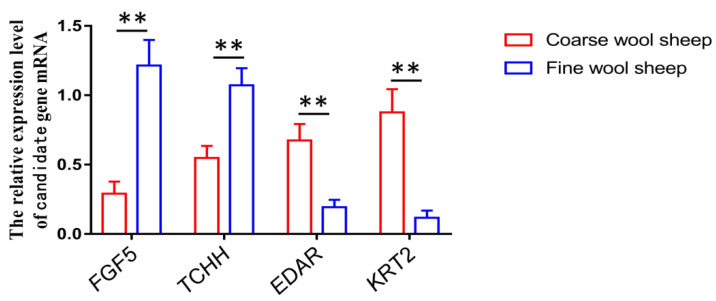
The Q-PCR results of four candidate genes between coarse and fine wool sheep skin samples, ** represents *p* < 0.01.

**Figure 8 biology-11-01637-f008:**
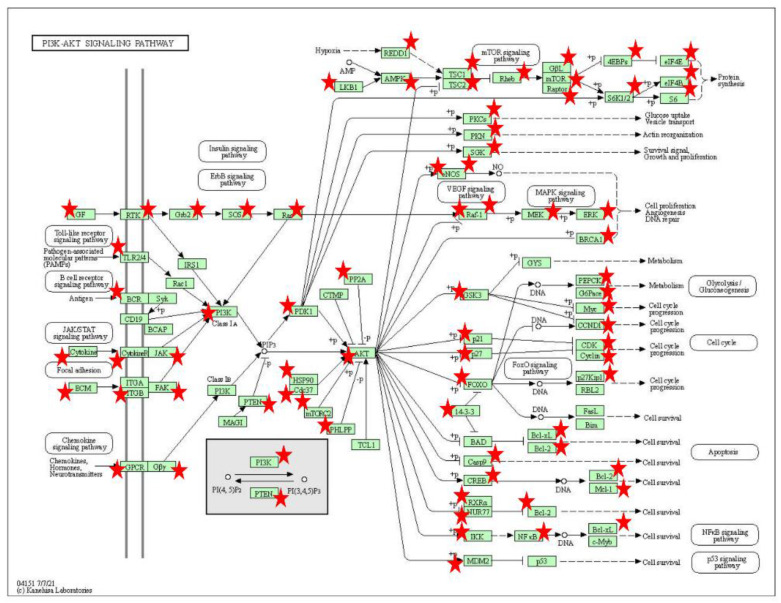
The PI3K/AKT signaling pathway (from KEGG).

**Table 1 biology-11-01637-t001:** Reads quality control. Q20% means that the sequencing error rate was less than 0.01, Q30% means sequencing error rate was less than 0.001.

Sample	Raw_Reads	Clean_Reads	Clean%	Q20 (%)	Q30 (%)	GC (%)
Coarse_IP	65,836,752	60,538,940	66.83	100	98.95	54.28
Coarse_input	59,483,600	52,447,544	75.24	100	99.25	53.92
Fine_IP	65,814,440	61,402,346	65.06	100	98.9	54.76
Fine_input	66,350,948	59,261,076	74.39	100	99.1	54.45

**Table 2 biology-11-01637-t002:** Clean reads mapping to the sheep reference genome.

Sample	Total Reads	Total Mapped (%)	Unique (%)	Non-Unique (%)
Coarse_IP	52,622,230	50,790,053 (96.52)	47,142,268 (92.82)	3,647,785 (7.18)
Coarse_Input	45,974,696	44,149,378 (96.03)	40,487,767 (91.71)	3,661,611 (8.29)
Fine_IP	53,580,892	51,594,011 (96.29)	48,063,658 (93.16)	3,530,353 (6.84)
Fine_Input	51,915,058	49,857,573 (96.04)	45,846,904 (91.96)	4,010,669 (8.04)

## Data Availability

The raw data has been made publicly available. SRA accession: PRJNA760832.

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
