# Peer review of "m6A Methylation Analysis Reveals Networks and Key Genes Underlying the Coarse and Fine Wool Traits in a Full-sib Merino Family"

_biology, 2022, doi:10.3390/biology11111637_

Round 1
Reviewer 1 Report
On the whole, the study appears very thorough and logical. I believe the results will be of interest to readership and are valuable and help to elucidate the molecular mechanisms involved in wool production. In my opinion, the manuscript should be accepted with minor corrections.
Reviewer 2 Report
General Comments (For Author)
This paper is considered to be an interesting paper on RNA-seq & epigenic analysis for the mechanism for wool quality in sheeps. In this study, I think that the authors showed an interesting and important topic. However, I think that the author should answer important questions in order to improve the completeness of the study.
Major Comments
1. (Introduction) I think the author needs to study in advance whether the trait for hair quality is a quantitative trait or a qualitative trait.
2. (Introduction) There are several sequencing methods related to epigenetics. Why did the author choose m6A Methylation Analysis?
3. (Method, Line 60) The traits were divided into two groups, I wonder what the exact criteria for dividing the two groups is.
4. (Method, Line 108) I wonder how the four genes for experimental verification were selected.
5. (Result, Line 179 & Figure05) RPKM is used for single-end sequencing.
6. (Result, Line 181) Of the 21,954 genes, 4,856 genes were statistically significant. Are too many genes statistically significant?
7. (Result, Figure05 C, D) m6A Peaks contained specific peaks. What do you want to show through this picture?
8. (Result, Line 219 & Figure07) What is the qpcr raw ct value? What is the result of comparing 2^-ddct and fc values ​​in fig 7?
9. (Discussion) The PI3K/AKT signaling pathway is not the only statistically significant result. Interpretation of other results is lacking. Author showed some selected genes from among many significant results, and it is necessary to prove that they play a central role among many genes (ex: network analysis).
Minor Comments
1. It is recommended to use commas when using 3 or more digits.
Reviewer 3 Report
The manuscript was the subject of sheep wool traits. It is interesting to note that coarse and fine wool traits was existed in a full-sib Merino family. Through m6A Methylation analysis, the researchers found PI3K/AKT pathway was participated in the formation of wool traits. It is pivotal for us to investigate the mechanisms of morphogenesis and development of wool follicles. After revision and scrutiny of the manuscript, there are some comments to argue with authors.
Comment 1: Please check the words spelling and figures’ notes.
Comment 2: Could you describe the traits more detailly? Would the trait change as sheep grows up?
Comment 3: Would you have some further plans about these associated genes and pathways revealed by your research? For revealing m6A mechanisms of primary wool follicle, would you plan to do the experiment which is the cross between coarse and fine sheep?
Comment 4: How could you apply your findings to the sheep breeding in the future? We are looking forward to some thoughts from you.
Round 2
Reviewer 2 Report
The author answered all the questions well.